# Metabolic Priming as a Tool in Redox and Mitochondrial Theragnostics

**DOI:** 10.3390/antiox12051072

**Published:** 2023-05-10

**Authors:** Sónia A. Pinho, Sandra I. Anjo, Teresa Cunha-Oliveira

**Affiliations:** 1CNC-Center for Neuroscience and Cell Biology, CIBB-Centre for Innovative Biomedicine and Biotechnology, University of Coimbra, 3060-197 Cantanhede, Portugal; spinho@cnc.uc.pt (S.A.P.); sandra.anjo@uc.pt (S.I.A.); 2PDBEB—PhD Programme in Experimental Biology and Biomedicine, Institute of Interdisciplinary Research (IIIUC), University of Coimbra, 3004-504 Coimbra, Portugal; 3IIIUC, University of Coimbra, 3004-504 Coimbra, Portugal

**Keywords:** metabolic priming, theragnostics, mitochondria, redox homeostasis, media composition, oxygen levels

## Abstract

Theragnostics is a promising approach that integrates diagnostics and therapeutics into a single personalized strategy. To conduct effective theragnostic studies, it is essential to create an in vitro environment that accurately reflects the in vivo conditions. In this review, we discuss the importance of redox homeostasis and mitochondrial function in the context of personalized theragnostic approaches. Cells have several ways to respond to metabolic stress, including changes in protein localization, density, and degradation, which can promote cell survival. However, disruption of redox homeostasis can lead to oxidative stress and cellular damage, which are implicated in various diseases. Models of oxidative stress and mitochondrial dysfunction should be developed in metabolically conditioned cells to explore the underlying mechanisms of diseases and develop new therapies. By choosing an appropriate cellular model, adjusting cell culture conditions and validating the cellular model, it is possible to identify the most promising therapeutic options and tailor treatments to individual patients. Overall, we highlight the importance of precise and individualized approaches in theragnostics and the need to develop accurate in vitro models that reflect the in vivo conditions.

## 1. Introduction

Antioxidants have been extensively studied in vitro due to their potential therapeutic benefits in alleviating oxidative stress and inflammation, which are linked to various diseases, such as cancer, cardiovascular diseases, and neurodegenerative disorders [1,2]. However, the translation of these findings into clinical practice has been challenging, as many clinical trials have failed to demonstrate significant benefits [3].

One reason for this discrepancy between in vitro and clinical outcomes could be the lack of translational potential of preclinical studies, which are often performed in controlled laboratory environments using animal models or cell cultures that do not accurately reflect the complexity of human physiology and disease processes. This can lead to over-optimistic predictions of therapeutic efficacy and safety [4,5].

More sophisticated models that can better mimic the complexity of human physiology and disease are needed to enhance the translational potential of preclinical studies. Another approach to enhancing the translational potential of preclinical studies is to conduct more rigorous and standardized research with well-defined study designs and end-points. This would improve the reliability and reproducibility of preclinical studies and enable more accurate predictions.

Metabolic priming, a technique that involves preconditioning cultured cells by manipulating cellular metabolism, has emerged as a valuable tool in theragnostics. This technique has significant potential to impact the study of cellular processes related to energy metabolism, particularly those associated with redox homeostasis and oxidative stress, which are believed to play a critical role in the development and progression of neurodegenerative diseases. This review explores the benefits of metabolic priming and its potential to advance our understanding of these complex diseases.

## 2. Mitochondrial Alterations in Health and Disease

Mitochondrial alterations can give rise to a wide array of diseases, including neurodegenerative disorders, metabolic disorders, and cancer. These alterations are often caused by mitochondrial stress and mitochondrial dysfunction, both of which will be discussed in this section.

### 2.1. Mitochondria as Dynamic Cellular Information Hubs

Mitochondria are renowned for their role as the powerhouses of the cell, responsible for producing most of the cellular energy in the form of adenosine triphosphate (ATP) through the process of oxidative phosphorylation (OXPHOS) [6]. However, mitochondria are involved in many processes beyond energy production. Mitochondria play a critical role in determining cell function and fate, playing roles in different cell processes, including, but not limited to, cell division, differentiation, calcium (Ca^2+^) signaling, autophagy, and apoptosis. Therefore, while their function as the cell’s powerhouse is undoubtedly significant, it is only one of many crucial roles that mitochondria play in maintaining cellular homeostasis [6,7,8]. Recognizing the diverse and multifaceted roles of mitochondria in cellular processes has led to the proposal of a new conceptualization of these organelles as central hubs for intracellular signal processing and integration. It is now understood that mitochondria are sophisticated information processing centers that are capable of sensing and responding to a wide range of signals from both within and outside the cell. This ability to integrate and respond to multiple signals makes mitochondria critical regulators of cellular function and highlights their importance in maintaining cellular homeostasis [9,10,11]. This new paradigm recognizes the complex interplay between mitochondrial function and various cellular processes, including metabolism, gene expression, and cellular signaling. Mitochondria play a critical role in regulating cellular behavior and maintaining cellular homeostasis by serving as hubs for integrating these signals. The updated conceptualization of mitochondria as an information processing system provides a more nuanced and comprehensive understanding of the functions of these organelles, emphasizing their central role in the complex web of cellular interactions that regulate cellular physiology and behavior [9].

Mitochondria are distinct cellular organelles with their own DNA (mtDNA) and a double-membrane system consisting of the mitochondrial outer (MOM) and inner membrane (MIM). Mitochondria also engage in extensive communication with other organelles. For example, they cooperate with peroxisomes in the catabolism of fatty acids through β-oxidation, share fission machinery components, and participate in redox signaling relationships [12]. Additionally, mitochondria interact with the endoplasmic reticulum (ER) in the regulation of Ca^2+^ homeostasis, lipid metabolism, and mitochondrial fission and autophagy [13], as well as with lysosomes in the regulation of mitophagy and fission [14]. Furthermore, mitochondria communicate with the Golgi apparatus in regulating mitochondrial dynamics [15] and with the nucleus for regulating cellular metabolism [15,16,17]. This communication underscores the social nature of mitochondria and their critical role in coordinating cellular processes. This organelle crosstalk is a key component of mitochondrial signaling in physiology and pathology.

Mitochondria are highly dynamic organelles moving in the cell by taking advantage of the microtubule network and dynamically changing their morphology by fission and fusion processes through specialized protein machinery that includes cytosolic dynamin-related protein 1 (DRP1) for fission and mitofusin (MFN1/2) and optic atrophy 1 (OPA1) for fusion at the MOM and MIM, respectively [18]. Cristae are also very dynamic structures whose shape and bioenergetic capacity change with physiological conditions, such as substrate availability, glucose, or oxygen (O_2_) deprivation [19,20]. The remodeling of cristae structure and OXPHOS function has been associated with alterations in cellular metabolism [19,21].

Recent findings using a powerful new tool called MitCOM have shown that the organization and interaction of mitochondrial proteins is more intricate than previously assumed [22]. MitCOM is based on high-resolution complexome profiling, which allows for the quantification of mitochondrial protein assemblies in the yeast mitochondrial proteome. This approach has provided new insight into mitochondrial protein machinery’s dynamics and molecular composition, revealing a more complex picture of mitochondrial function and organization [22]. The mitochondrial respiratory chain (MRC), also known as the electron transport chain (ETC), consists of enzymes that form supramolecular structures called supercomplexes (SCs). This evidence contradicts the traditional fluid model of dispersed enzymes in the MIM and in cristae membranes [23,24]. Complexes I, III, and IV can assemble into one SC known as the respirasome [23,25]. Recent evidence has shown the co-existence of two independent MRC structures: the C-MRC, which is bioenergetically more efficient under oxidative metabolic conditions, and the S-MRC, which emerges when metabolic reconfiguration favors glycolysis. These structures have been described in human skin fibroblasts and the HEK 293T and osteosarcoma 143-B cell lines, as well as postmitotic tissues, such as the brain frontal cortex and skeletal muscle [26]. The two structures are regulated by three isoforms of the mitochondrial respiratory SC assembly—stabilizing factor, COX7A (COX7A1/2), and SC assembly factor 1 (SCAF1, also known as COX7RP and COX7A2L)), as well as by the state of the pyruvate dehydrogenase complex (PDH), which in its active form promotes OXPHOS and the C-MRC structure. When PDH is inactive, representing a metabolic shift to glycolysis, the S-MRC structure is promoted. This important discovery emphasizes the link between metabolic reconfiguration and the MRC architecture, involving SC reorganization in response to environmental signals, such as variations in the metabolic conditions, increased oxidative stress, or the instability of individual complexes [25,26]. Cytochrome c oxidase (COX), also known as complex IV, is composed of 13 different subunits encoded by both nuclear and mitochondrial genomes and catalyzes the transfer of electrons from cytochrome c to molecular O_2_, generating water as a byproduct. COX activity is fine-tuned in response to various physiological conditions, and this regulation may include the incorporation of different COX subunit isoforms produced through alternative splicing of nuclear-encoded subunit isoforms and/or post-translational modifications, including reversible phosphorylation. It can also occur through the binding of nucleotides, hormones, or proteins/enzymes and by the formation of SC with other complexes involved in OXPHOS [27,28]. Thus, mitochondrial dynamic activity is correlated with energy demand and nutrient supply, responding to bioenergetic adaptation.

Mitochondria have two mechanisms for controlling metabolism, homeostasis and adaptation, which primarily differ in how metabolic sensing and downstream effects are coupled. Homeostasis refers to the maintenance of a relatively stable internal environment, while adaptation refers to the ability to adjust to environmental changes. Mitochondria, as biosynthetic organelles with metabolically plastic and unique stress response processes, require both homeostasis and adaptation mechanisms to maintain optimal function. Two distinct circuits of metabolic control, feedforward and feedback control, are used to achieve metabolic homeostasis or metabolic adaptation in response to perturbations [10]. Feedback control circuits exert a direct effect on the sensed metabolic parameter retroactively to counteract perturbations via a negative feedback mechanism, while feedforward control mechanisms rewire the metabolic network to ensure adequate metabolic output in the presence of perturbation [10]. In some scenarios, these two circuits act simultaneously, engaging a downstream effector that both maintains homeostatic control and promotes an adaptive response [10]. The related concept of mitohormesis, and its relationship with metabolic priming, is described in Section 4.

Maintaining a proper balance of several mitochondrial processes, including biogenesis, dynamics, proteostasis, mitophagy, Ca^2+^ homeostasis, and redox signaling, is essential for the maintenance of healthy mitochondria [7,8,29]. These processes work together to create mitochondrial quality control mechanisms, ensuring that dysfunctional mitochondria are eliminated and healthy ones are maintained. However, when these balances are disrupted, mitochondria can become dysfunctional, producing an excess of reactive redox species (RRS) that surpass the capacity of the cellular antioxidant system to detoxify them, leading to oxidative stress [29].

Cellular metabolism relies on a highly interconnected and well-coordinated network of chemical reactions, involving constant intra- and extracellular communication among cell organelles. These reactions organize into several metabolic pathways, such as glycolysis, citric acid cycle (also known as tricarboxylic acid) (TCA), pentose phosphate pathway (PPP), OXPHOS, or fatty acid β-oxidation. Metabolites act as important intermediate molecules and substrates for these biological processes, but some metabolites also function as signaling molecules that can influence the activity of regulatory proteins, nutrient sensing, cell survival and differentiation, and embryonic development, being key regulators of cell phenotype and behavior [29,30]. Mitochondrial metabolites such as acetyl-coenzyme A (ACCoA) can act as second messengers, inducing post-translational modifications that modulate the activity of metabolic enzymes [30,31]. The balance between catabolic and anabolic pathways is crucial to maintaining metabolite levels and ensuring an adequate energy supply. Therefore, cells constantly adapt to different metabolic needs and conditions of cellular stress to maintain normal cellular physiology, relying on mitochondrial plasticity, which plays a crucial role in cellular and mitochondrial homeostasis [30].

### 2.2. Mitochondrial Stress

The term “mitochondrial stress” refers to the state of the mitochondria when they are exposed to various stressors that can disrupt their normal functions, such as oxidative stress, nutrient deprivation, mitochondrial DNA damage, and other factors [31,32]. Mitochondrial stress can trigger various cellular responses, including changes in mitochondrial morphology, altered mitochondrial function, and the induction of mitochondrial quality control mechanisms. These responses are critical for maintaining cellular homeostasis and preventing the accumulation of damaged mitochondria, which can lead to cellular dysfunction and disease [33,34] (Table 1). On the other hand, mitohormesis is a biological response where the induction of a reduced amount of mitochondrial stress leads to an increment in health and viability within a cell, tissue, or organism [35,36], which can lead to persistent adaptations that protect the mitochondria and the cell against subsequent stressors [37]. Mitonuclear communication is crucial for coordinating information exchange between the nuclear and mitochondrial stress responses when triggered by potentially harmful stimuli. Among the signals, the most important for hermetic response in mitochondria are reactive species (such as superoxide (O_2_^.−^) or hydrogen peroxide (H_2_O_2_)), mitochondrial metabolites, proteotoxic signals, mitochondrial to cytosol stress response, release of cytokines (specifically myokine), insulin/IGF-1 receptors, AMP-activated protein kinase (AMPK) and mammalian target of rapamycin (mTOR), sirtuins, and unfolded protein response (UPR) [36,38].

### 2.3. Mitochondrial Dysfunction

Mitochondrial dysfunction can be triggered by internal (genetics/mutations) or external (drug toxicity) causes and is implicated in several diseases, namely, cancer and metabolic and neurodegenerative disorders, among other pathologies [30,64]. Cellular and mitochondrial alterations depend on the type of disease (Table 1).

In cancer cells, an increase in oxidative stress can derive from mutations on ETC complexes, resulting in increased DNA damage, variations of the number and integrity of mtDNA copies per cell, inadequate repair due to the low repair capacity of mitochondrial DNA, mtDNA mutations, and chromosomal abnormalities and instability [39,40,41]. Cancer cells also suffer mitochondrial dysfunction [42] and often metabolic switch from OXPHOS to glycolysis leading to lactate accumulation, which is now considered an important regulator of cancer growth and an active signaling molecule [43,44,45,46,47,48]. In metabolic disorders, such as type 2 diabetes (T2D), obesity, and NAFLD, the main alterations are associated with cell metabolism [53,54]. An excess of nutrient supply and physical inactivity increase the risk of insulin resistance and T2D, contributing to an increase in mitochondrial H_2_O_2_ and general oxidative stress levels due to an overload of TCA and ETC or an increase in inflammatory processes [49,50]. The abnormal changes in circulating fuel levels and in how central and peripheral tissues use energy substrates also result in mitochondrial dysfunction throughout several organ systems [58]. Since mitochondria play a crucial role in cell metabolism, researchers have been developing mitochondria-targeted molecules. This subject is discussed in more detail in Section 4.4.2, Mitochondria-Targeted Drugs [52].

Neurodegenerative diseases also have an important component of mitochondrial dysfunction and loss of redox homeostasis, as described in Table 1 [55,56,57,58,59,60,64]. Thus, mitochondria are important drug targets for neurodegenerative diseases, and some small molecules or peptide sequences targeting mitochondria are being developed (see Section 4.4.2) for different mitochondrial targets, since several mitochondrial structures/functions are affected in these diseases [65].

In addition, mitochondrial dysfunction impacts not only various pathologies but also the abundance and thermal stability of cellular proteins. This has been demonstrated through multidimensional analysis of pre–post thermal proteome profiling (ppTPP) using isobaric peptide tags in combination with pulsed stable isotope labeling by amino acids in cell culture (pulsed SILAC). This analysis allows for the monitoring of time-sensitive adaptations of mature (pre) and newly synthesized proteins (post) to specific insult-inducing stress, allowing the study of mitoprotein-induced stress responses [66].

Exposure to pharmacological drugs and other chemicals can also induce toxicity involving adverse mitochondrial effects in different cells and tissues depending on the type of drug, which can cause mitochondrial dysfunction through multiple mechanisms. Mitochondrial adverse effects of drugs are a common reason for the withdrawal of many drugs from the market, representing expensive failures [67,68]. Several drugs, such as troglitazone for T2D, cerivastatin for hyperlipidemia and cardiovascular disease prevention, and N-phenethylbiguanide for diabetes, have been withdrawn from the market due to their associated toxicity, including hepatotoxicity and rhabdomyolysis leading to renal failure. Additionally, nefazodone, an antidepressant/anxiety drug, was also withdrawn from the market due to its hepatotoxicity associated with mitochondrial damage [67,69,70]. Notably, all of these drugs have been linked to mitochondrial damage as the underlying mechanism of their toxicity.

In summary, mitochondrial alterations and dysfunction have been implicated in various diseases. Understanding the role and mechanisms of mitochondrial alterations in disease pathogenesis may lead to developing novel therapies that target mitochondrial dysfunction to prevent or treat these diseases.

### 2.4. Mitochondrial Function as an Important End-Point in Drug Development

As previously mentioned, mitochondrial toxicity is a severe concern in drug development, but it can be predicted through preclinical studies involving mitochondrial safety screenings [71], which are essential to identify mitochondrial liabilities in the early stages of drug development [72]. Understanding harmful drug–mitochondrial interactions could aid in creating safer treatment plans for individual patients, which is important in precision medicine [67], a topic that is covered in more detail in Section 4.4. However, compounds’ complete range of effects on mitochondrial function cannot be assessed without standardized systematic techniques properly designed and validated to screen and characterize them. This is necessary to determine their safety, therapeutic windows, and potential applications. In fact, even candidate mitochondrial protectants must be screened for potential mitochondrial liabilities to determine their safe concentration windows.

## 3. Mitochondrial Stress Leading to Cellular Redox and Metabolic Remodeling

### 3.1. Cellular Responses to Metabolic Stress

Cells respond to metabolic stress to ensure their homeostasis and survival (Figure 1). Such responses include (1) activation of stress response pathways, such as the AMPK and mTOR pathways, which regulate cellular processes, including energy production, metabolism, and protein synthesis [73,74]; (2) alteration in energy production, which enables cells to switch to alternative metabolic pathways, such as fatty acids or amino acids as fuel sources, or decrease their energy expenditure to conserve resources [75]; (3) the upregulation of autophagy, where the cellular process of degrading and recycling cellular components removes damaged organelles and decreases stress [76]; (4) activation of antioxidant pathways and repair mechanisms to prevent or reverse oxidative damage caused by increased RRS production [75]; or (5) triggering of immune responses, which promotes tissue repair and clearance of damaged or dead cells and tissues [77].

Cell cycle and intracellular metabolism are closely interconnected [75], so metabolic stress also affects cell proliferation. Cell division demands a substantial energy and biomass synthesis, making it highly reliant on the metabolic state and nutrient availability, which influence the cell’s decision to start dividing [78]. In fact, metabolic enzymes control cell cycle checkpoints. For example, aerobic glycolysis actively supports protein synthesis, especially in the G1 phase of the cell cycle, whereas dimeric pyruvate kinase M2 (PKM2), a glycolytic enzyme, promotes cellular biosynthesis and the expression of cyclin D1. Another glycolytic enzyme, 6-phosphofructo-2-kinase/fructose-2,6-biphosphatase 3 (PFKFB3), increases ATP production at the G1 restriction checkpoint and promotes the activation of cyclins D and E. The upregulation of glutaminase-1 (GLS1) enhances glutaminolysis, which sustains DNA replication during the S phase. Moreover, glycolytic enzymes, such as PKM2, aldolase, fructose-bisphosphate (ALDO), and glyceraldehyde 3-phosphate dehydrogenase (GAPDH), can translocate to the nucleus, where they upregulate the expression of genes involved in the cell cycle [79]. Thus, it is unsurprising that energy substrates and metabolites highly influence cell cycle progression. For example, the accumulation of lactate due to elevated glucose catabolism in highly proliferative cells has recently been shown to influence cell proliferation by altering the activity of the anaphase-promoting complex/cyclosome (APC/C) through the direct inhibition of the SUMO protease sentrin-specific protease 1 (SENP1). Specifically, lactate-mediated inhibition of SENP1 leads to the stabilization of SUMOylation on two residues of anaphase-promoting complex subunit 4 (APC4), resulting in the degradation of cell cycle proteins, such as kinases and cyclins. While lactate accumulation signals that cells are in a nutrient-rich environment and stimulates cell division, excessive lactate can lead to aberrant cell division [80]. These recent discoveries have improved our understanding of the link between metabolism and cellular behavior in various contexts [81] and have important implications for ex vivo and in vitro assays using proliferative cells.

Maintaining cellular function relies on the delicate balance between ATP synthesis and hydrolysis. During metabolic stress, such as the loss of mitochondrial transmembrane potential (MMP), a component of the proton motive force, ATP hydrolysis is triggered through the reversal of the mitochondrial ATP synthase (also known as complex V) activity. The mitochondrial protein ATPase inhibitor IF1 can inhibit this reverse enzymatic activity. Recent research in isolated heart mouse mitochondria and in primary cultures of fibroblasts obtained from patients has found that selective inhibitors of this reverse enzymatic activity can ameliorate mitochondrial respiration and restore cellular homeostasis in mitochondrial pathologies [82]. The researchers identified (+)-epicatechin, a polyphenol, as a selective inhibitor of ATP hydrolysis that binds to ATP synthase without affecting its ATP synthesis activity. In fact, inhibiting the hydrolytic activity of ATP synthase in cells with complex-III deficiency was sufficient to restore ATP content without re-establishing respiratory function [83]. Moreover, (+)-epicatechin was shown to enhance muscle force in a mouse model of Duchenne muscular dystrophy by decreasing ATP hydrolysis [82].

Various stress signals, including hypoxia, stress hormones, and high levels of glutamate or glucose, can elevate cytosolic Ca^2+^ concentration. This leads to the activation of a mitochondrial phosphatase that dephosphorylates COX, resulting in the loss of its allosteric ATP inhibition. Consequently, there is an increase in MMP and ROS formation. Excessive ROS production can cause apoptosis or various diseases [84]. Additionally, inflammatory signaling can hinder OXPHOS by phosphorylating the COX catalytic subunit I at Tyr304. This results in a decrease in MMP and energy production, which can lead to organ failure and death, as observed in septic patients [85].

Deficiencies in the succinate dehydrogenase complex (SDH), complex II, can induce metabolic stress and stimulate cellular metabolic remodeling. SDH is responsible for supporting the synthesis of aspartate, which is essential for cellular proliferation. However, unlike other impairments in the ETC, supplementation with electron acceptors failed to alleviate the effects of SDH inhibition. Recent research has shown that the co-inhibition of ETC complex I can restore aspartate production and promote cell proliferation in SDH-deficient cells [86]. This restoration depended on the decrease in mitochondrial NAD^+^/NADH through pyruvate carboxylation and glutamine reductive carboxylation, highlighting the impact of mitochondrial redox changes on cellular fitness [86].

COX plays a central role in the regulation of OXPHOS, being the final enzyme in the ETC and contributing to the generation of the proton motive force required for ATP synthesis. As previously mentioned, COX activity is tightly regulated by multiple factors and, thus, the dysregulation of COX activity has been implicated in a range of disease states, including neurodegenerative disorders, cancer, and mitochondrial diseases. Therefore, it is crucial to understand the intricate regulation of COX for the maintenance of cellular homeostasis and prevention of diseases [87,88].

Interactions between proteins and metabolites play a role in the dynamic and tissue-specific metabolic flexibility that allows for growth and survival in response to constantly changing nutrient environments. A new platform was recently developed to enable the discovery of protein–metabolite interactions based on mass spectrometry integrated with equilibrium dialysis for the discovery of allostery systematically (MIDAS) with greater sensitivity than previous methods. By using this technique, researchers have shown that exposure of H9c2 rat cardiomyoblasts to fatty acids has effects in metabolic stress that cause a decrease in pyruvate–lactate interconversion, revealing the interconnected regulation of lactate dehydrogenase (LDH) with carbohydrate metabolism [89]. Interestingly, ATP and long-chain fatty acyl-CoAs inhibit the isoform A of lactate dehydrogenase (LDHA) at physiological concentrations, highlighting the importance of protein–metabolite interactions in regulating metabolic pathways [89].

### 3.2. Protein Distribution and Density in Response to Metabolic Stress

In response to metabolic stress, cells can change the distribution and density of specific proteins to conserve energy, maintain homeostasis, and improve their survival chances. These changes can occur in various organelles and cell locations, including the mitochondria. The transport of proteins across cellular membranes presents several challenges, mainly due to their size and chemical heterogeneity [90,91]. Firstly, during or after translation at cytoplasmic ribosomes, proteins must be precisely directed to their intended destinations and inserted into or crossed through lipid bilayers without disrupting membrane integrity, a process known as protein translocation. Secondly, mitochondrial-specialized machinery, such as the mitochondrial intermembrane space import and assembly (MIA) machinery, is required for regulating protein translocation to adapt to dynamic cellular demands [92]. In addition to maintaining the proper balance between mitochondrial and nuclear genomes, cells must also detect and compensate for any mutations or conditions that may disrupt the activities of translocation systems. These systems ensure that mitochondrial proteins are properly transported and assembled within the organelle. An imbalance in the expression or function of these systems can lead to defects in mitochondrial protein homeostasis and ultimately impair mitochondrial function. To avoid such outcomes, cells have developed sophisticated surveillance mechanisms that monitor mitochondrial protein import and assembly and can initiate compensatory responses, such as the upregulation of chaperones and proteases to ensure proper mitochondrial function [91,93]. To ensure proper protein folding and repair, molecular chaperones such as heat-shock protein 70 (Hsp70), TRAP1, and chaperonin Hsp60/10 complexes, are essential. Mitochondrial proteostasis also relies on specific proteases to degrade irreparable protein damage [93], essential for mitochondrial quality control, including mitophagy. Additionally, the mitochondrial unfolded protein response (UPRmt), which is a mitochondria stress response, is transcriptionally activated by mitochondrial chaperone proteins [94], being the transcription of target UPRmt genes epigenetically regulated by histone 3-specific methylation [3]. Interestingly, UPRmt delays aging and extends the lifespan by promoting mitochondrial proteostasis [95], and this process is regulated through activating transcription factor 1 (ATFS-1) in *Caenorhabditis elegans* and its mammalian homologue activating transcription factor 5 (ATF5) [94]. In mammals, the integrated stress response (ISR) and UPRmt are strictly associated, being potential drug targets in the treatment of age-related diseases, such as Alzheimer’s disease (AD), Parkinson’s disease (PD), Huntington’s disease (HD), and amyotrophic lateral sclerosis (ALS) [3,96]. Moreover, a study utilizing multiple omics techniques has found that the activating transcription factor 4 (ATF4) plays a crucial role in regulating the response of mitochondria to stress in mammalian cells [97]. Mitonuclear responses are activated upon mitochondrial stress to preserve and restore mitochondrial function and improve the adaptation to cellular stress and ATF4 stimulates the expression of cytoprotective genes which, together with the activation of ISR, will reprogram cellular metabolism [97].

Mitochondrial proteins can undergo redistribution in response to metabolic stress caused by factors such as nutrient deprivation, hypoxia, oxidative stress, and exposure to toxins or drugs. This redistribution can originate from alterations in gene expression derived from transcriptional and post-transcriptional regulation. For example, the transcriptional co-activator peroxisome proliferator-activated receptor gamma coactivator 1-alpha (PGC-1α) is activated in response to metabolic stress to enhance the expression of genes involved in mitochondrial metabolism and biogenesis [98]. MicroRNAs can also target specific mitochondrial proteins during metabolic stress. One example is miR-210, which downregulates the expression of the key enzyme cytochrome c oxidase (COX, or complex IV) during hypoxia [99], while miR-383 targets mitochondrial peroxiredoxin 3, which is involved in reactive oxygen species (ROS) detoxification and apoptosis, during hyperglycemic circumstances [100]. The translocation of nuclear-encoded miRNAs to mitochondria is possibly associated with mitohormesis [101]. While an extensive discussion of this topic is beyond the scope of the current manuscript, it has previously been addressed in the published literature [102,103,104]. ATFS-1 transcription factor is translocated from mitochondria to the nucleus to regulate UPRmt genes, which may happen because ATFS-1 has a weak mitochondrial targeting sequence that allows for dynamic subcellular localization during initial UPRmt activation [33,105].

Post-translational modifications, such as phosphorylation, acetylation, and ubiquitination, can also change the function of mitochondrial proteins during metabolic stress. For instance, AMPK, activated during energy stress, can phosphorylate and activate enzymes involved in mitochondrial biogenesis and metabolism, such as PGC-1α and the mitochondrial transcription factor (TFAM) [106,107]. Some post-translational modifications can also cause changes in cristae morphology, such as cristae narrowing [108]. The assembly of SCs is stabilized by many factors, including the (SCAF1, and the respiratory SC factor 1/2 (RCF1 and RCF2)) in mice liver mitochondria [109] and in breast and endometrial cancer cells under hypoxia [110]. OPA1 is an important cristae modulator that responds to cellular alterations through a mechanism dependent on mitochondrial carrier family SLC25A proteins [21,111]. Moreover, mutations in the mitochondrial-encoded ND1-ND5 complex I subunits, involved in SC formation, can impair mitochondrial function in cholestatic liver disease, suggesting a connection between SCs alteration and metabolic diseases [63]. In addition, some mitochondrial proteins, such as MFN2, glucose-regulated protein 75 (GRP75), and protein tyrosine phosphatase-interacting protein 51 (PTPIP51) for Ca^2+^ homeostasis, phosphofurin acidic cluster sorting protein 2 (PACS-2) which participates in the trafficking of ion channels, can cluster around the mitochondria-associated membranes (MAMs) during nutrient deprivation to facilitate energy and Ca^2+^ transfer from the ER in response to changes in cellular homeostasis [112,113,114]. The modifications in the expression and function of mitochondrial proteins during metabolic stress are complex and multidimensional, involving several regulatory processes that help to adjust mitochondrial metabolism and maintain cellular energy homeostasis [115,116]. Thus, UPRmt has a role in mitochondrial quality control being activated when mitochondrial integrity and function are compromised [95,117]. It can also sense defects in mitochondrial translation and deal with proteotoxic stress through two signaling pathways: UPRmt and unfolded protein response activated by the mistargeting of proteins (UPRam) [117].

During the normal aging process and in individuals with diabetes, a persistent hexokinase 1 (HK1) relocation from the mitochondria to the cytosol has been described. This shift is associated with increased production of inflammatory markers, including the cytokines interleukin IL-1b and IL-6 and TNFalpha, and a decrease in GAPDH protein activity. As a result, an increase in PPP metabolic flux was observed. The location of HK1 within the cell influences the metabolic fate of glucose [118]. Moreover, under metabolic stress, the mitochondrial sentrin/SUMO-specific protease (SENP) family (SENP2) controls the assembly of SDH. In response to glutamine deprivation, SENP2 slows down mitochondrial respiration. This stress condition selectively activates SENP2, leading to the deSUMOylation of the SDHA subunit and a subsequent decrease in the activity of SDH complex [119].

Proteins residing in the ER can also undergo redistribution in response to metabolic stress, activating the UPR. Normally, 78 kDa glucose-regulated protein (GRP78) binds to the sensor proteins inositol-requiring enzyme type 1 (IRE1), eukaryotic translation initiation factor 2 alpha kinase 3 (EIF2AK3), also known as protein kinase R (PKR)-like endoplasmic reticulum kinase (PERK), and activating transcription factor 6 (ATF-6), which are located in the ER membrane. However, when cells are under stress, these sensor proteins can be released from GRP78, initiating a cascade of signal transduction that activates either survival or death pathways [120,121,122,123]. ER stress causes the ER to move closer to the mitochondria and activate bioenergetics through the PERK-ERO1α complex, which controls mitochondrial dynamics by promoting the oxidation of mitochondria-ER contact sites (MERC) proteins [124]. The combination of oxireductase Ero1α and PERK prevent energy depletion and oxidative stress in the ER and cytosol by oxidizing important signaling proteins upon MERCs [124]. Both the formation of mitochondrial SCs and bioenergetics are stimulated in response to ER stress and glucose deprivation through the PERK axis, which activates SCAF1 expression [13]. The effects of Ca^2+^ and sodium on the control of MIM fluidity and OXPHOS activity support the idea that the super-assembly plays a crucial role in modifying the mitochondria’s metabolic response [125].

Furthermore, transcription factors can translocate from the cytoplasm to the nucleus to activate genes involved in stress response pathways. These include hypoxia-inducible factor 1-alpha (HIF-1α), which is activated under hypoxic conditions [126,127], or nuclear factor erythroid 2-like 2 (NFE2L2/Nrf2), which regulates redox homeostasis and several cytoprotective mechanisms that confer adaptation to stress conditions. Nrf2 binds to Kelch-like ECH-associated protein 1 (Keap1) in the cytoplasm, and, under stress, it is released and translocated to the nucleus to activate the transcription of cytoprotective and metabolic genes [46]. Nuclear translocation of mitochondrial proteins can also occur upon cellular stress, as the coordination between the nucleus and mitochondria is critical for normal cell development [128]. For example, pyruvate dehydrogenase kinase isoform 2 (PDK2) translocates from the mitochondria to the nucleus in different mouse and human xenograft tumor models of prostate cancer [129], and a heterogeneous expression and subcellular localization of the PDH complex in human prostate carcinoma LNCaP cells was also observed [130]. PDH was also found to translocate to the nucleus in a cell-cycle-dependent manner in response to serum, epidermal growth factor, and mitochondrial stress, indicating the importance of this translocation for the local generation of ACCoA required for histone acetylation [131,132]. Moreover, the nuclear localization of other mitochondrial enzymes involved in the TCA cycle (including pyruvate carboxylase (PCB), aconitase 2 (ACO2), citrate synthase (CS), mitochondrial isocitrate dehydrogenase 3A (IDH3A), oxoglutarate dehydrogenase (OGDH), succinate dehydrogenase A (SDHA), and malate dehydrogenase 2 (MDH2)) can modulate histone acetylation [132]. In fact, mitochondrial function has an important role in epigenetic regulation and DNA damage response, indicating that cells also respond to epigenetic alterations during environmental stress [133].

In addition to protein location changes, protein density and degradation can also occur in response to metabolic stress. For example, cells can decrease the overall density of ribosomal proteins under nutrient deprivation to diminish the energy cost of protein translation [134]. Cells can also upregulate the degradation of specific proteins to conserve energy or remove misfolded or damaged proteins [135,136], namely, through the ubiquitin–proteasome system, preventing their accumulation and toxicity [137].

In summary, the proper distribution of proteins between nucleus and mitochondria is essential for maintaining cellular homeostasis and regulating various cellular processes, such as energy production, DNA replication, and protein synthesis. Abnormal protein distribution or defects in the protein import machinery between both structures have been linked to various aging-related diseases, including neurodegenerative disorders, cancer, and metabolic syndromes [128]. Furthermore, changes in protein localization, density, and degradation can also occur in response to metabolic stress, helping cells to conserve energy, maintain homeostasis, and promote cell survival. For instance, studies have shown that the accumulation of nuclear-encoded mitochondrial proteins in the cytosol can cause mitochondrial dysfunction and oxidative stress, which are key contributors to aging and age-related diseases. Further research is needed to understand the underlying mechanisms better and develop potential therapeutic strategies to target protein distribution in the treatment of age-related diseases [128,138,139,140,141,142].

### 3.3. Cellular Redox Alterations

Cellular redox homeostasis relies on a crucial and dynamic system that controls a wide range of biological responses and ensures the balance between reducing and oxidizing reactions within cells. To maintain redox homeostasis, cells need to continuously adapt, identify changes in the redox status, and restore redox equilibrium when it is disturbed. Alterations in redox homeostasis are associated with the progression of aging [143,144,145], representing one of the major health concerns nowadays, with implications in diseases. Reactive redox species, such as ROS, reactive nitrogen species (RNS), and reactive sulfide species (RSS), play a crucial role in normal cell activity (metabolism), stress responses and inflammation. However, during oxidative stress, redox equilibrium is disturbed, impairing cellular functions, and increasing the risk of diseases [146,147,148]. To fully understand the mechanisms of redox control, it is necessary to have a comprehensive knowledge of the intricate chemistry underlying reactive species and their interactions with both electron donors and acceptors. The cells rely on an efficient antioxidant defense system to maintain the redox balance and prevent and repair cell damage caused by excessive reactive species levels. This defense includes both enzymatic systems, such as ascorbate peroxidase, catalase, glutathione peroxidase (GPX), glutathione reductase, glutathione-S-transferase, superoxide dismutase, and peroxiredoxin, as well as nonenzymatic systems, such as ascorbate, glutathione (GSH) and tocopherol, which act against an excess of oxidants. Antioxidant drugs have been suggested to be beneficial for aging-related diseases, as this type of molecules is part of the cellular defense system. However, despite promising in vitro assays, most clinical trials with antioxidants have failed, yielding contradictory results and even showing harmful effects [149]. The limitations of clinical trials with antioxidants indicate a potential issue with the design of preclinical assays for these agents. This may be due to inappropriate preclinical biomarkers or targets (e.g., mitochondria or redox-sensitive proteins), incorrect biological models, or errors in analytical methods (e.g., the use of unvalidated or inappropriate methods, loss of information). It is also important to consider that antioxidant agents can interfere with normal intracellular reactive species signaling, highlighting the need for caution when designing new redox therapies. Recently, the “5R” principles of precision redox pharmacology were proposed as guidelines for redox pharmacology, emphasizing the importance of considering individual differences in redox levels and physiological conditions when selecting the right species, place, time, level, and target for intervention [149,150]. This is important since oxidative stress markers usually evaluate DNA, proteins, or oxidative lipid damage or DNA damage and repair, which are late end-points and may only represent cells that have already passed the point of no return. Early biomarkers that detect the first mechanisms of action and may enable stopping oxidative stress as soon as possible should be more effective [150], and must consider the interplay/crosstalk between oxidative stress response and cell metabolism. These could be a basis for better screening tests for selecting the most appropriate molecule for a specific application [151].

### 3.4. Redox In Vitro Models

Mitochondrial failure resulting in oxidative stress and cell damage underscores the importance of exploring the underlying mechanisms of diseases and developing new therapies. Leveraging metabolic preconditioning by subjecting cells to changes in O_2_ and nutrient availability in the culture medium when developing models of oxidative stress and mitochondrial dysfunction may facilitate this process. Furthermore, challenging patient cells ex vivo with oxidant agents, such as peroxides, can be an interesting approach to test their redox status. By forcing the antioxidant system to respond, this approach can reveal any subclinical disease manifestations and may help to identify new therapeutic targets [152].

Several chemical inducers of oxidative stress, such as H_2_O_2_ [153], menadione [153,154] and tert-butyl hydroperoxide (t-BHP) [155,156] have been used in in vitro assays to disturb cellular redox homeostasis. Testing more than one agent with different mechanisms of action can help reveal deficits in different redox pathways. For instance, H_2_O_2_ is a stable, diffusible molecule that can penetrate biological membranes and react with intracellular targets. It is a nonradical ROS produced as a byproduct of cell metabolism, and can directly damage DNA, lipids, and other macromolecules, leading to cellular oxidative injury [153,154]. GPX and catalase metabolize H_2_O_2_ intracellularly [157,158], but if not metabolized, it can react with transition metals such as ferrous ion Fe^2+^ or copper ion Cu^2+^, resulting in the generation of the highly reactive hydroxyl radical (^•^OH) that can quickly react with biomolecules [159], often on the order of microseconds to milliseconds, near the diffusion limit (>10^9^ M^−1^ s^−1^) [160]. H_2_O_2_ can modify protein activity by reacting with thiol groups of cysteine residues to form sulfenic acid, and disulfide bonds, leading to protein conformational changes [157,158]. In addition, H_2_O_2_ can activate redox-sensitive transcription factors, such as Nrf2 and activator protein 1 (AP-1), leading to changes in gene expression and triggering the cellular response to oxidative stress [46].

Another well-known oxidative stress inducer is t-BHP, a lipid-soluble organic peroxide that can cross the plasma membrane and accumulate in the cytosol. t-BHP can undergo intracellular metabolization, either enzymatically via GPX or cytochrome P450, or nonenzymatically by interacting with intracellular reducing agents such as GSH or ascorbic acid [161]. Metal ions catalyze the decomposition of t-BHP into alkoxyl and peroxyl radicals, and this can generate RRS, including H_2_O_2_ and O_2_^.-^. Metabolization by cytochrome P450 promotes lipid peroxidation, DNA damage, depletion of cellular GSH and protein thiols, disruption of intracellular Ca^2+^ homeostasis, cell damage, and apoptosis [161,162].

Although the exact mechanisms by which ROS can signal metabolic stress and regulate various cellular processes such as metabolism, proliferation, inflammation, and cell death are not fully understood, a new working model has been proposed to describe the integrated network by which ROS systems sense and respond to changes in homeostasis and stress across different subcellular localizations (including mitochondria, peroxisomes, plasma membrane, cytosol, and nucleus). This model suggests that targeting ROS signaling pathways may hold great promise for treating different conditions, including cardiovascular diseases, inflammatory disorders, and tumors [163].

## 4. Metabolic Priming In Vitro/Ex Vivo for Mitochondrial Theragnostics

In order to improve translational research (Figure 2), several actions should be taken into account during theragnostic studies: (a) selecting an appropriate cellular model that reflects the disease or condition being studied is crucial [164]; (b) preconditioning cells by adjusting cell culture conditions, and manipulating cellular metabolism through changes in pH, O_2_ levels, and nutrient availability is essential to simulate the original cellular environment as closely as possible [165,166,167,168]; (c) introducing cell genetic modifications to enhance their suitability for theragnostic applications [169,170]; (d) validating the cellular model is crucial to ensure its appropriateness for a specific purpose or context of use [171]. This validation process may include analyses based on morphology (shape, size, and organization of the cells, using machine learning techniques could be helpful [172]), function (e.g., response of cells to specific stimuli [165,167,168,171,173]), and genotype (expression of specific genes or the existence of genetic mutations [165,174]).

Metabolic priming strategies applied to a wide range of pathologies can be achieved through hypoxia, nutrient deprivation, or metabolite supplementation, which will be discussed further in the following subsections. These manipulations can create cellular environments resulting from the interactions between different cell types, which may represent both physiological and pathological tissue conditions [175,176,177,178]. Various cellular models have been used, including in vitro monolayers like fibroblasts, which are easily obtainable from patients using minimally invasive methods and carry disease-relevant cellular dysfunction [165,173], two- dimensional (2D) and three-dimensional (3D) ex vivo models (Figure 2). Examples of those models are human neuron/astrocyte coculture and histocultures (2D and 3D), and organoids derived from human induced pluripotent stem cells (iPSCs), mesenchymal stem cells (MSCs) or spheroid cultures (3D). These can be used for cell type-specific manipulations or personalized drug discovery, with the advantage of more closely mimicking the original tissue architecture. In addition, extracellular matrix modulation by changing its stiffness also influences several phenotypic features of stem cells through mechanotransduction [52,179,180]. These models have been used for personalized drug discovery and to study disease-relevant cellular dysfunctions [166,176,177,178,181,182,183,184,185,186,187,188,189,190] (Figure 2).

### 4.1. Media Composition

Cell culture medium is a liquid nutrient solution designed to provide the essential nutrients for cell growth, such as vitamins, amino acids, inorganic salts, glucose, serum-derived growth factors and hormones [175]. It plays a critical role in supporting cell growth and proliferation during in vitro culture. Initially, cell culture media, such as Dulbecco’s Modified Eagle Medium (DMEM), were developed with the aim of promoting high cell proliferative capacity, without taking into account the impact on cellular metabolism studies [71,175,191]. However, as our understanding of cellular metabolism has grown, concerns have been raised about the impact of culture medium on cell physiology. Researchers are now adjusting in vitro conditions to match real-world conditions and taking into account the specific needs of each cell type based on the purpose of the study. New formulations were developed to provide more physiological significance, such as human plasma-like medium (HPLM) [192] and Plasmax [191], which are sold as a ready-to-use liquid solution. As an alternative, the medium can be prepared using a powder including vitamins, amino acids, and mineral salts, with the remaining components added in the correct amounts for the cells in study. The added components can include different energetic substrates, sources of carbon or buffers, such as glucose, galactose, glutamine, glutamate, sodium pyruvate, HEPES, or sodium bicarbonate. Therefore, selecting the appropriate culture medium is crucial for developing in vitro culture models. However, optimizing the cell culture medium can be a time-consuming and arduous process due to the vast number of possible combinations [175]. One way to precondition cells using culture medium is by altering substrate availability. The choice of substrate can impact the energy source for the cells. For example, switching from glucose to galactose in the medium can change the cells’ reliance on glycolysis versus OXPHOS for energy production [193]. It should be noted that in cases where galactose is the carbon source, the cells may rely on glutamine as an energy source. The ability of cells to adapt to changes in their environment is critical for their survival and function. This change in the culture media can be immediate [152] or gradual [165,194,195], and these two strategies may be valuable for different purposes. When transitioning to a new culture medium containing different energy substrates, cells must quickly switch from one mode of energy production to another, which can pose a significant challenge. A rapid change in substrate availability can cause metabolic stress and impair cellular function, especially in cells with limited metabolic flexibility, such as cancer cells. On the other hand, a gradual change in the culture medium, with a decrease in the concentration of the previous substrates and an increase in the new ones, allows for a gentler molecular reconfiguration, enabling cells to adapt to the new metabolic conditions without compromising their viability or functionality. Optimizing the protocol used is crucial as some cell lines, particularly cancer-derived ones, may have difficulty adapting quickly to a glucose-free culture medium.

Commercially available media typically contain 25 mM glucose (Figure 3), which is supra-physiological glucose concentration (HGm), creating a hyperglycemic environment. Under these conditions, cells tend to rely heavily on glycolysis for energy production, rather than on mitochondrial respiration, which can affect the study of cellular metabolism, particularly mitochondrial metabolism. Consequently, the use of HGm can lead to inaccurate experimental results and hinder the understanding of cellular physiology. Therefore, researchers must carefully select the appropriate glucose concentration in their culture medium to ensure that the cellular metabolic pathways are accurately reflected, especially when studying mitochondrial function [196,197]. To maintain a more natural and physiologically relevant environment for cellular metabolism studies, glucose concentration in the culture medium should be at physiological levels, typically approximately 5 mM. This is achieved by using a low glucose medium (LGm). In cases where glucose needs to be absent, glutamine-modified media can be used, where glucose is replaced by galactose as the carbon source. This is commonly referred to as galactose-containing OXPHOS-stimulating medium (OXPHOSm), which promotes a shift to mitochondrial respiration and OXPHOS, and is particularly useful for studying cellular energy production and metabolism [193,198]. This shift leads to an increase in mitochondrial mass and activity, as well as a rewiring of the redox homeostasis. In human skin fibroblasts, a gradual adaptation from HGm to LGm or OXPHOSm has been shown to cause this rewiring [165]. The use of OXPHOSm has been suggested as a more precise method for assessing mitochondrial toxicity and has been successfully utilized for predicting drug-induced liver injury [165]. In opposition, commonly used HGm can mask mitochondrial toxicities that would be relevant under more physiological conditions [198,199,200,201,202,203]. To comprehensively understand the mechanisms involved in testing different drugs or developing an oxidative in vitro model to test antioxidant molecules, all three media (i.e., HGm, LGm and OXPHOSm) can be used in parallel. This approach enables researchers to determine whether the effects of antioxidants or other drugs are related to mitochondrial activity. Using all three media can also help to identify the cellular mechanisms involved in oxidative stress and how cells adapt to different energy substrates, thereby contributing to a better understanding of cellular metabolism and improving drug development processes [71]. As discussed previously, it is essential to carefully control cell culture conditions to accurately detect mitochondrial toxicity in vitro (Figure 3).

### 4.2. O_2_ Levels

O_2_ is involved in many metabolic reactions, some sensitive to variations in that gas (Figure 3). However, physiologically relevant O_2_ levels in cell culture experiments have not been used in most in vitro experiments [204]. In fact, conventional cell culture conditions are far from mimicking the in vivo environment, with the cells being cultured at atmospheric O_2_ levels, near 18–19%, which corresponds to an hyperoxic condition [204,205]. These levels correspond to at least 10 times higher levels of O_2_ than required to sustain maximal mitochondrial respiration rates [204]. Most mammalian cells in an organism never experience such high levels of O_2_. In vivo, physiological O_2_ levels (physioxia) range from 1 to 11% depending on the tissue perfusion and cell type [204,205,206,207]. Additionally, fluctuations in O_2_ levels have been associated with numerous pathophysiological conditions, including cancer. Hypoxia is considered a hallmark of cancer and is thought to be related to cancer development and malignancy [205,206,207,208]. Variations in O_2_ levels can significantly impact on mitochondrial activity and redox homeostasis, which rely on O_2_ as a crucial mediator [206]. The immediate effect of exposing cells to a hyperoxic environment in standard cell culture is the disruption of redox homeostasis caused by increased production of ROS and RNS from specific O_2_-consuming organelles and enzymes, including mitochondria, NADPH oxidases, and other sources such as the transplasma membrane redox system, nitric oxide synthases, xanthine oxidase, and monoamine oxidase [204,207]. These phenotypic alterations are well-established to occur under hyperoxic conditions. This will potentially lead to more oxidative and nitrosative damage in cells. DNA strand breaks, DNA and RNA oxidation, protein carbonyl levels, and lipid peroxidation are among some of the oxidative and nitrosative already reported to be increased in hyperoxic conditions compared to culturing the cells under physioxia [204,207]. Furthermore, ROS/RNS elevation triggers redox-sensitive and inflammatory signaling pathways by activating redox-sensitive transcription factors and other signaling molecules. The central pathway induced by low-sustained ROS level involves the induction of the antioxidant and cellular detoxification program associated with activating the Nrf2/Keap1 regulatory pathway. When activated, Nrf2 will bind the antioxidant response elements encoding antioxidant enzymes, including those involved in GSH metabolism, and involved in the generation of reducing equivalents [206,207]. Finally, at more intermediate amounts of ROS, factor nuclear kappa B (NF-κB), activators of transcription (STATs), and AP-1 transcription factors are activated, as well as some mitogen-activated protein kinases (MAPKs). Altogether, this will induce a vast transcription regulation that will promote various downstream processes, including inflammation, survival, and apoptosis [206,207].

Contrarily, lower levels of O_2_ lead to the stabilization of HIF1-α and consequent activation of the HIF pathway. Initially suggested as a mechanism of response to hypoxia, later data also suggest that HIF1 activation may also have a role in physiological O_2_ conditions [209]. This evidence is associated with the fact that the O_2_ sensors that mediate HIF1-alpha stabilization, prolyl-hydroxylases (PHD1-3), and the asparaginyl-hydroxylase factor inhibiting HIF1 (FIH) are susceptible to O_2_ changes and, thus, can be gradually activated in response to alterations in O_2_ levels in the physio-normoxic range, resulting in a graded transcriptional response appropriate to the O_2_ levels sensed. HIFs transcriptionally regulate hundreds of genes; most of the O_2_-consuming and ROS/RNS-producing enzymes described above are among these targets. In addition to regulating antioxidant enzymes and enzymes involved in GSH metabolism, HIFs also regulate several aquaporins that facilitate H_2_O_2_ diffusion within cells. This suggests that O_2_ levels may also impact intercellular communication by altering cellular membrane permeability and the diffusion of H_2_O_2_. Furthermore, HIF1 targets several enzymes involved in glucose metabolism, including PDK, phosphofructokinase, and LDH, indicating that O_2_ levels also play a role in the regulation of cellular metabolism [204,207,210,211]. In fact, broad O_2_-dependent effects on cellular metabolism and mitochondrial function were observed in physio-normoxia (or hyperoxia) comparative studies. It is worth noting that different outcomes have been observed depending on the cell type, which may be attributed to the specific and unique metabolic profiles of each cell type. Further details on this aspect are reviewed in depth in [204]. Nonetheless, some general trends have been observed, including a widespread increase in glycolysis and decrease in OXPHOS in physioxia, as well as a decrease in mitochondrial metabolic activity and an increased OCR/extracellular acidification rate (ECAR) ratio (oxidative metabolism) in cells cultured at higher levels of O_2_ ([204] and references cited therein). In addition, mitochondrial dynamics is affected by O_2_ levels. Although only few studies have addressed this aspect, it is possible to conceive that, in general, physioxia tends to promote mitochondrial fusion (more elongated mitochondria), while hyperoxia increases mitochondrial fission (rounder mitochondria and smaller network size) ([204] and references therein). Indeed, several studies have provided evidence that culturing cells under atmospheric conditions can cause significant alterations in the mitochondrial proteome and structure, leading to changes in mitochondrial function, particularly related to oxidative respiration. For instance, one study found that culturing human embryonic stem cells under atmospheric O_2_ conditions resulted in altered mitochondrial morphology and the upregulation of stress response pathways [20,212]. Another study observed that culturing cells under high O_2_ levels led to mitochondrial fragmentation and decreased OXPHOS, which was associated with decreased levels of mitochondrial fission protein Drp1 [20]. Additionally, studies have confirmed that culturing cells at atmospheric O_2_ levels resulted in alterations to the mitochondrial morphology, decreased ATP production, and increased ROS production [209].

In summary, changes in O_2_ levels during cell culture, especially when cells are cultured in hyperoxic conditions (atmospheric O_2_ levels), have been shown to dysregulate various signaling pathways and cellular processes. In hyperoxic environments, cells are constantly exposed to higher levels of oxidative stress and have lower energy production capacity, leading to altered responses to external stimuli, such as drugs, hormones, and toxins. This phenomenon has been confirmed by several studies ([204] and references therein). In line with this, physiological O_2_ conditions facilitate the maintenance of in vivo-like properties of cells cultured in vitro [204]. Taken together, the various pieces of evidence discussed above underscore the importance of carefully controlling culture O_2_ levels to obtain meaningful results from in vitro studies. The dysregulation of signaling pathways and cellular processes, along with the constitutive exposure to oxidative stress and lower energy production observed in hyperoxic conditions, can significantly alter the response of cells to external agents, including drugs, hormones, and toxicants. Given the potential implications of these findings for disease pathology and therapeutic development, it is crucial to prioritize O_2_-level control in in vitro studies to maximize their relevance and translational potential [209].

### 4.3. Evaluation Tools for Mitochondrial Health Status

Mitochondrial health plays a crucial role in maintaining the overall vitality of the entire body, as it mediates cellular function across various tissues, both in healthy and diseased states. Quantifying these statuses can facilitate evaluating a patient’s health. Thus, a healthy mitochondrial profile can be defined by functional markers, such as OCR and MMP, morphological markers such as the extent of mitochondrial network and fusion/fission status, as well as biochemical/metabolic markers, including protein levels, mRNA levels, mtDNA copy number, metabolite levels, substrate consumption, enzyme activities, and free radical levels and production [213]. Two potential systems for assessing mitochondrial health have been proposed: Mitochondrial Health Index (MHI) and Bioenergetic Health Index (BHI).

The MHI combines measures of mitochondrial content and functional capacity, incorporating a mathematical combination of the respiratory chain enzymatic activity containing nuclear and mitochondrial DNA-encoded subunits (a marker of energy production capacity) and the mtDNA copy number (a marker of mitochondrial mass). This is expressed as:$$MHI=\frac{energy\;production\;capacity}{mitochondrial\;content\times 100}$$

A low MHI was found in isolated peripheral blood mononuclear cells (PBMCs) of chronically stressed caregivers, specifically mothers of children with autism spectrum disorders, suggesting that daily mood and chronic caregiving stress affects mitochondrial health [214]. Moreover, in patients who have bipolar disorder (BD), a low MHI, also measured in isolated PBMCs, was found to be associated with high plasma circulating cell-free mitochondrial DNA reflecting a negative clinical result [215]. By using the MHI, both studies suggested a link between mitochondrial health, psychological stress, and disease pathophysiology [214,215,216].

As alterations in bioenergetic pathways are associated with various metabolic diseases, including diabetes, cancer, and cardiovascular and neurodegenerative diseases (see Table 1 and Section 2.3 for details), the BHI has been proposed as a potential biomarker for assessing patient health, having prognostic and diagnostic values from the perspective of personalized cell-based assays and translational research [216,217,218]. The BHI relies on a mitochondrial assay based on the OCR. A high BHI is associated with high bioenergetic reserve capacity, high ATP-liked respiration, and low proton leak, and it can be calculated using the following equation:$$BHI=\frac{ATP-linked\;OCR\times reserve\;capacity}{proton\;leak\;OCR\times non-mitochondrial\;OCR}$$

The BHI has also been used to measure the susceptibility of cells isolated from patients to oxidants generated by the redox cycling agent 2,3 dimethoxynaphthoquinone (DMNQ), providing bioenergetic profiles of monocytes from healthy human subjects [217,218].

### 4.4. How Mitochondria Can Be Used as a Biosensor for Drug Development

#### 4.4.1. Redox Biomarkers

Redox biomarkers are molecules or cellular processes that reflect the balance between oxidation and reduction reactions within cells and tissues [219]. These biomarkers can identify normal and pathological processes or reactions to therapeutic interventions [220]. The literature on the subject of redox biomarkers is vast and covers various changes that can occur following exposure to oxidative stress or different types of diseases, which can alter biological markers. Given the scope of this topic, we focus only on some of the redox biomarkers that have been investigated in different diseases.

For example, salivary GSH and 4-hydroxynonenal (4-HNE), products of lipid peroxidation, can be potential biomarkers to measure the degree of systemic oxidative stress in patients with insulin resistance, although additional validation is needed to confirm their diagnostic efficacy [221]. Advanced oxidation protein products (AOPPs), oxidized free cysteine, tumor necrosis factor receptor-1 (TNFR1), and osteopontin are promising biomarkers associated with the loss of estimated glomerular filtration rate (eGFR) in immunoglobulin A (IgA) nephropathy, a kidney disease also known as Berger’s disease. IgA nephropathy occurs when IgA deposits accumulate, causing inflammation and damage to kidney tissues [222]. Furthermore, high plasma levels of the heme-scavenger alpha-1-microglobulin (A1M) were found in women with high-risk preeclampsia, a relatively common hypertensive disorder in pregnancy, suggesting that plasma A1M may be a biomarker for illnesses associated with pregnancy [223]. A1M was also suggested as a potential biomarker to assess long-term risk of developing arthrosis, namely, acute inflammatory arthritis and osteoarthritis, following kidney injuries [224]. Moreover, significantly higher levels of malondialdehyde (MDA) and decreased GPX activity were found in the plasma of patients with heart failure when compared to the control group, with a positive correlation between the levels of MDA and the duration of the heart failure, suggesting MDA and GPX as potential biomarkers for that disease [225].

The investigation of the effects of UV-induced oxidative stress on human skin has been facilitated by the development of a new tool based on electron spin resonance spectroscopy and two-dimensional ultra-weak photon emission. This tool can detect the formation of triplet-excited carbonyls and singlet O_2_ in skin biopsies, providing a more precise understanding of the mechanisms underlying oxidative damage [226]. In addition, transthyretin, an extracellular protein responsible for transporting thyroxine and retinol-binding protein complex to various parts of the organism, has been implicated in cardiovascular failures, polyneuropathy, psychological disorders, obesity, and diabetes [226,227,228,229]. Its function has been associated with oxidative stress [227,228], and it is a potential biomarker for oxidative stress-related illnesses, such as PD and AD [228,229,230]. Other possible oxidative stress biomarkers for neurodegenerative diseases include markers of lipid oxidation, such as 4-HNE and F2-isoprostanes, markers of protein oxidation, such as protein carbonylation and oxidation of glutamyl residues, markers of nucleic acid oxidation, such as 8-hydroxy-2-deoxyguanosine (8-OHdG) and 8-hydroxyguanosine, and markers of glycation, such as advanced glycation end products (AGEs) [230]. Increased levels of 8-OHdG and MDA in different types of cancer were also proposed as important markers by defining the stage of tumor progression [231]. Overall, redox biomarkers provide useful information on the redox status of cells and tissues and can be used to track the development of diseases, find novel drug targets, and assess the success of therapeutic interventions.

Developing new drugs is challenging due to the complex nature of biological systems, lengthy process involved, and dependence on multiple factors. Additionally, the heterogeneity of patient populations, lack of validated diagnostic and therapeutic targets, and ambiguity in study design add to the high cost of drug development [232,233,234]. However, science is constantly evolving and exploring new approaches to enhance drug discovery, including multi-omics. This method integrates different omics, such as genomics, transcriptomics, proteomics, metabolomics, and proteogenomics, to analyze and interpret the underlying mechanisms involved in biological processes [235,236]. Additionally, artificial intelligence, such as machine/deep learning algorithms are being used to handle large and complex datasets in drug discovery [236,237]. These innovative approaches have the potential to accelerate drug discovery and improve the efficiency of drug development.

One promising area of research in drug development is targeting redox homeostasis, a vital process for normal cell development. For instance, Nrf2 has emerged as a promising therapeutic target in cancer cells [46,238], neurodegenerative diseases [239,240], and ischemic stroke [241]. Together with Nrf2, class O forkhead box transcription factor (FoxO) activators or NF-κB inhibitors have been used for schizophrenia [242] and ischemic stroke [241,242].

#### 4.4.2. Mitochondria-Targeted Drugs

Mitochondria-targeted antioxidants have gained attention due to the crucial role of mitochondria in redox homeostasis. Triphenylphosphonium cation (TPP^+^)-based mitochondria-targeted antioxidants, such as MitoQ, SkQ1, AntioOxCIN4, and AntOxBEN2 (also known as MitoBEN2), have been developed. MitoQ, based on coenzyme Q, has been demonstrated to activate the Nrf2 signaling pathway and protect intestinal epithelial cells from mtDNA damage induced by ischemia/reperfusion or hypoxia/reoxygenation in both in vitro and in vivo studies [243]. AntiOxCIN4, derived from the conjugation of caffeic acid with an alkyl linker and TPP^+^, has been shown to improve liver steatosis in Western diet-fed mice [52], revert cellular and mitochondrial abnormalities in male sporadic Parkinson’s disease patients’ skin fibroblasts [244], and stimulate ROS-protective pathways in primary human skin fibroblasts by inducing Nrf2 and increasing superoxide dismutase 2 (SOD2) and GSH levels [245]. AntiOxBEN2, which is derived from hydroxybenzoic acid conjugated with TPP^+^, presented promising results in delaying aging by preventing senescence in skin cells [246], indicating that both AntiOxCIN4 and AntiOxBEN2 may act as pro-oxidants, triggering internal ROS defense mechanisms [245]. SkQ1, a mitochondria-targeting derivative of plastoquinone and TPP^+^, has been used in preclinical studies for the treatment of cardiovascular, renal, and inflammatory bowel diseases in an intestinal epithelium mice model and Caco-2 cells [247]. In addition, SS-31, a mitochondria-targeting peptide, which has an amino acid sequence that enables it to cross plasma membrane independently of MMP and localize to MIM, was shown to decrease mitochondrial ROS in isolated heart mitochondria. Recent studies have also highlighted the importance of targeting mitochondria in cancer cells from the perspective of mitochondria drug development. Strategies include MitoTam in breast cancer [49,50,51] or class 5 mitocans, which include compounds that target the ETC [248], among other drugs [52,249]. In summary, these drugs that specifically target mitochondria are promising strategies for counteracting oxidative stress in mitochondria.

#### 4.4.3. Mitochondrial Theragnostics

The concept of theragnostics has been around since the early days of nuclear medicine when radioactive compounds were used for diagnostic imaging and radionuclide therapy in metastatic thyroid cancer [250,251]. Today, theragnostics has evolved into a more precise and individualized approach that combines diagnostics and therapeutics into a single integrated approach, being successfully applied in various medical fields, including oncology [252], neurology [253], cardiovascular [253,254], and infectious diseases [255]. Incorporating this approach into clinical practice relies on ex vivo studies using the patient’s biological material, and metabolic priming strategies can be important allies to achieve the desired models. By modulating the cellular environment in a way that can accurately model the in vivo state of cells obtained from patients, a metabolic priming can mimic the physiological and pathological conditions to a better extent, thus helping diagnose subclinical defects (gnostics) and test therapeutic strategies (thera) without subjecting the patient to experimental treatments. Moreover, it is possible to study changes in metabolism, gene expression, signaling pathways; identify new drug targets; and screen drugs under specific metabolic conditions [253,254,256]. Therefore, this will potentiate the creation of patient-specific disease models, identify personalized treatment options, and predict treatment outcomes based on the patient’s metabolic profile [253].

Several studies have explored using OXPHOSm for studies in patient skin fibroblasts. One such study, conducted approximately 30 years ago, found that fibroblasts from patients with oxidative deficiencies did not survive when grown in galactose medium, while fibroblasts from patients with hepatic form of COX deficiency and mild PDH complex deficiency were able to survive. This suggests that culturing cells in galactose medium may serve as a screening test for patients with oxidative defects [257], although it is not clear if the medium contained glutamine to feed the mitochondria. Recently, OXPHOSm was used to challenge human skin fibroblasts obtained from sporadic Parkinson’s disease (sPD) patients and force them to use mitochondria to survive, allowing to uncover subclinical metabolic defects of fibroblasts from sPD patients that were otherwise concealed [174], and demonstrating the importance of this strategy for mitochondrial theragnostics. Another study utilized OXPHOSm to investigate the biochemical consequences of mitochondrial dysfunction in complex-I- and PDH-deficient patients and found that preconditioning cells can aid in distinguishing between normal and metabolic deficient cells. Furthermore, untargeted NMR analysis can be employed to differentiate cellular metabolic adaptations in PDH and complex-I-deficient fibroblasts [258].

Finally, human primary muscle cells cultured in glucose-free medium containing galactose (and glutamine) were used to investigate the effect of galactose (and glucose deprivation) on oxidative metabolism in post-diabetic patients. The results showed that this medium enhanced the oxidative metabolism and revealed mitochondrial dysfunction-related mechanisms [199]. This approach to metabolic priming-associated theragnostics can offer great possibilities for developing new therapies, screening drug efficacy, and advancing personalized medicine. By using preconditioning to mimic the real-life organism as closely as possible, researchers can study disease mechanisms and identify potential therapeutic targets (see Figure 2). Furthermore, by validating the cellular models, including morphological, functional, and genotypic analyses, the appropriateness of the models for specific purposes or contexts of use is assured [167,168].

## 5. Conclusions

Aging and related diseases are linked to alterations in the cellular redox equilibrium, presenting opportunities for developing precision redox medicine strategies. Despite the promising benefits of antioxidants from in vitro studies, clinical trials have often failed to produce the desired outcomes. Efforts must be made to enhance the translational potential and efficacy of preclinical studies and theragnostics. One way to achieve this is by establishing in vitro models that closely mimic the physiological and pathological conditions of the organism, and this involves selecting appropriate cell models and tailoring culture conditions to mimic in vivo environments, such as glucose and O_2_ levels and tissue rigidity. Metabolic priming is a useful tool for achieving this goal, as it helps to mitigate the impact of artificial redox environments in conventional cell culture conditions on mitochondria regulation, leading to a deeper understanding of cellular redox responses. Moreover, metabolic priming can enable the accurate diagnosis of subclinical disease manifestations and personalized screening of various therapeutic strategies without risking patient harm. However, this is still a challenge, as there is a lack of technology that allows for the use of some metabolic priming strategies in a high-throughput manner, limited understanding of some of these priming methods, and a lack of information on combining multiple types of priming to fully address the in vivo cellular milieu. Overall, the use of metabolic priming is a valuable strategy for theragnostics, but further research is needed to optimize its use and expand its potential applications.

## Figures and Tables

**Figure 1 antioxidants-12-01072-f001:**
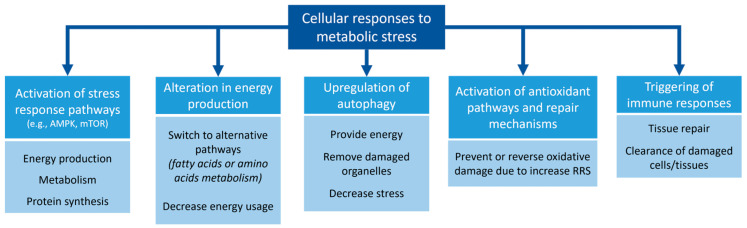
Different cellular responses to metabolic stress and their consequences. Metabolic stress can occur in cells when there is an imbalance between energy demand and supply, disrupting cellular homeostasis. Different types of metabolic stress can occur, including glucose deprivation, hypoxia, oxidative stress, and nutrient excess. Cells respond to metabolic stress by activating various signaling pathways and metabolic pathways to restore homeostasis. For more details see Section 3.1.

**Figure 2 antioxidants-12-01072-f002:**
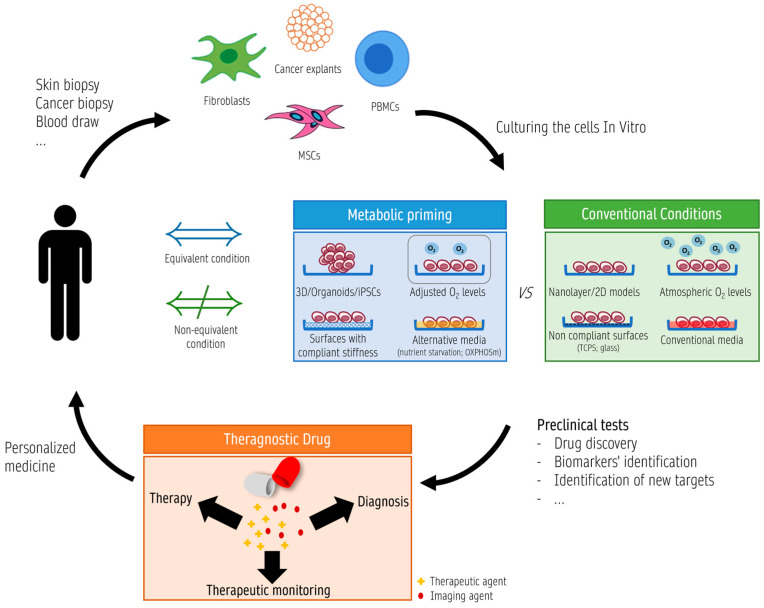
Overview of the potential application of metabolic priming in translational research and theragnostics. Primary cells (such as fibroblasts, PBMCs, and others) collected from patients and control individuals can be cultured in vitro to serve as good cellular models to study disease pathophysiology and to test new therapeutic drugs. Different strategies of metabolic priming (such as adjusted O_2_ levels, OXPHOS stimulation, and others) have emerged as an alternative to conventional culturing conditions, creating an in vitro environment that closely reflects the conditions of the cells in the organism. By improving the translational potential of the findings obtained in the in vitro/ex vivo studies, the preconditioned (metabolically primed) cells can be a valuable tool in theragnostics. For more details, see Section 4.4.3.

**Figure 3 antioxidants-12-01072-f003:**
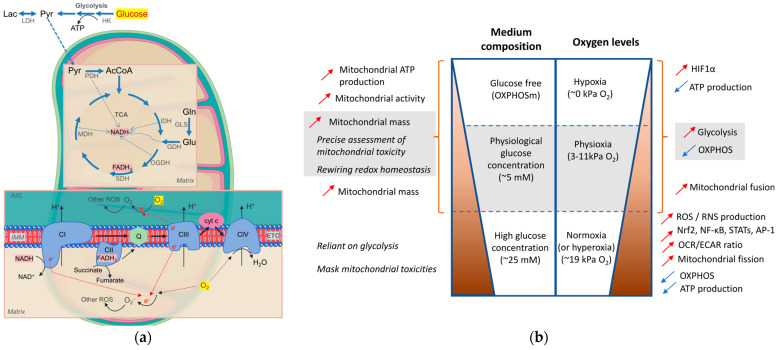
Cellular and mitochondrial rewiring in different metabolic conditions. (**a**) Interaction of glucose and oxygen (O_2_) with mitochondrial bioenergetics. In the cytosol, glucose is converted into pyruvate (Pyr), which is imported into mitochondria and converted into acetyl coenzyme A (AcCoA). AcCoA enters the tricarboxylic acid cycle (TCA) following a series of reactions that produce reducing equivalents, such as NADH and FADH_2_. Electrons from redox equivalents are transported through the electron transport chain (ETC) in reactions coupled with the export of protons (H^+^) that contribute to the establishment of a pH and an electric gradient, resulting in a proton motive force that fuels the conversion of ADP into ATP by ATP synthase (not represented). Electrons (e^−^) that leak from the ETC may partially reduce O_2_ into superoxide anion (O_2_^.−^), which may be converted into other more reactive oxygen species (ROS). GDH—glutamate dehydrogenase; Glu—glutamate; Gln—glutamine; GLS—glutaminase; HK—hexokinase; IDH—isocitrate dehydrogenase; IMM—inner mitochondrial membrane; IMS—intermembrane space; OGDH—oxoglutarate dehydrogenase; Lac—lactate; LDH—lactate dehydrogenase; MDH—malate dehydrogenase; PDH—pyruvate dehydrogenase; SDH—succinate dehydrogenase. Some elements used in this figure were obtained from smart.servier.com. (**b**) Metabolic remodeling induced by glucose (or its absence) and O_2_ levels and its impact on energy production and redox system. Culturing cells in medium containing different glucose concentrations, in the presence of glutamine and pyruvate, and exposure to different O_2_ levels have an impact on cellular responses. This emphasizes the importance of carefully controlling the cell culture medium and O_2_ levels to obtain meaningful results from in vitro studies. For more details, see Section 4, Section 4.1, and Section 4.2.

**Table 1 antioxidants-12-01072-t001:** Alterations of redox and metabolic homeostasis implicated in several diseases.

Disease	Redox- or Metabolic-Related Alterations	References
Cancer	Defects in mtDNA	[39,40,41]
	Metabolic shift from OXPHOS to glycolysis	[42]
	Increased mitochondrial fission	[43,44,45,46,47,48]
Metabolic disorders (T2D, obesity, NAFLD)	TCA cycle and mitochondrial respiratory chain overload	[49,50,51]
	Mitochondrial malfunction due to the accumulation of free fatty acids in adipose and peripheral tissues (lipotoxicity)	[49,50,51]
	Increased oxidative stress	[49,50,51]
	Insulin resistance	[49,50,51]
	Dysregulation of Ca^2+^ homeostasis	[50]
	Compromised mitochondrial bioenergetics	[52]
	mtDNA mutations	[53,54]
Neurodegenerative diseases (AD, PD, HD, ALS)	Impaired mitochondrial biogenesis and mitochondrial dynamics	[55,56,57]
	Excess ROS production	[58]
	Decreased intracellular Ca^2+^ buffering	[58]
	Decreased respiratory capacity and/or loss of mitochondrial transmembrane potential	[59]
	Disruption of intracellular trafficking-associated neurotoxicity	[52,60,61,62,63]

mtDNA—mitochondrial DNA; OXPHOS—oxidative phosphorylation; TCA—tricarboxylic acid cycle; ROS—reactive oxygen species; AD—Alzheimer’s disease, PD—Parkinson’s disease; HD—Huntington’s disease; ALS—amyotrophic lateral sclerosis.

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
