# Peer review of "Metabolic Priming as a Tool in Redox and Mitochondrial Theragnostics"

_antioxidants, 2023, doi:10.3390/antiox12051072_

Round 1

Reviewer 1 Report

From my point of view I congratulate to the authors for this very interesting review. It is easy to read and easy to understand. But with all respect I have to insist in 3 points so far.

1.      When to talk about isoforms and isoform regulation (page 3/120), then my friend Bernhard should be mentioned. It is only fare, isn’t it? Even in regard to the IF 14.

Kadenbach B, Hüttemann M. The subunit composition and function of mammalian cytochrome c oxidase. Mitochondrion. 2015 Sep;24:64-76. doi: 10.1016/j.mito.2015.07.002. Epub 2015 Jul 17. PMID: 26190566.

Especially the paper of Reguera DP et al. (Cells 2020; 9, 443) – COX 4I2- should also be mentioned

2.      Even when to talk about „mitochondrial stress” then I would appreciate to include some aspects from the Huttemann- and former Kadenbach- group.

Ramzan R, Vogt S, Kadenbach B. Stress-mediated generation of deleterious ROS in healthy individuals - role of cytochrome c oxidase. J Mol Med (Berl). 2020 May;98(5):651-657. doi: 10.1007/s00109-020-01905-y. Epub 2020 Apr 20. PMID: 32313986; PMCID: PMC7220878.

Lee I, Hüttemann M. Energy crisis: the role of oxidative phosphorylation in acute inflammation and sepsis. Biochim Biophys Acta. 2014 Sep;1842(9):1579-86. doi: 10.1016/j.bbadis.2014.05.031. Epub 2014 Jun 4. PMID: 24905734; PMCID: PMC4147665.

Kadenbach B, Ramzan R, Vogt S. High efficiency versus maximal performance--the cause of oxidative stress in eukaryotes: a hypothesis. Mitochondrion. 2013 Jan;13(1):1-6. doi: 10.1016/j.mito.2012.11.005. Epub 2012 Nov 23. PMID: 23178790.

Hüttemann M, Helling S, Sanderson TH, Sinkler C, Samavati L, Mahapatra G, Varughese A, Lu G, Liu J, Ramzan R, Vogt S, Grossman LI, Doan JW, Marcus K, Lee I. Regulation of mitochondrial respiration and apoptosis through cell signaling: cytochrome c oxidase and cytochrome c in ischemia/reperfusion injury and inflammation. Biochim Biophys Acta. 2012 Apr;1817(4):598-609. doi: 10.1016/j.bbabio.2011.07.001. Epub 2011 Jul 13. PMID: 21771582; PMCID: PMC3229836.

Kadenbach B, Ramzan R, Vogt S. Degenerative diseases, oxidative stress and cytochrome c oxidase function. Trends Mol Med. 2009 Apr;15(4):139-47. doi: 10.1016/j.molmed.2009.02.004. Epub 2009 Mar 18. PMID: 19303362.

Samavati L, Lee I, Mathes I, Lottspeich F, Hüttemann M. Tumor necrosis factor alpha inhibits oxidative phosphorylation through tyrosine phosphorylation at subunit I of cytochrome c oxidase. J Biol Chem. 2008 Jul 25;283(30):21134-44. doi: 10.1074/jbc.M801954200. Epub 2008 Jun 5. PMID: 18534980; PMCID: PMC3258931.

3.      For the regulation of OXPHOS the CytOx has a “center stage”. So, some more words about complex IV and interactions regarding respiration would be fine.

Arnold S. The power of life--cytochrome c oxidase takes center stage in metabolic control, cell signalling and survival. Mitochondrion. 2012 Jan;12(1):46-56. doi: 10.1016/j.mito.2011.05.003. Epub 2011 May 26. PMID: 21640202.

Ramzan R, Kadenbach B, Vogt S. Multiple Mechanisms Regulate Eukaryotic Cytochrome C Oxidase. Cells. 2021 Feb 28;10(3):514. doi: 10.3390/cells10030514. PMID: 33671025; PMCID: PMC7997345.

Helling S, Hüttemann M, Ramzan R, Kim SH, Lee I, Müller T, Langenfeld E, Meyer HE, Kadenbach B, Vogt S, Marcus K. Multiple phosphorylations of cytochrome c oxidase and their functions. Proteomics. 2012 Apr;12(7):950-9. doi: 10.1002/pmic.201100618. PMID: 22522801; PMCID: PMC3728716.

Author Response

Thank you for taking the time to review our manuscript and for your positive comments and insightful suggestions. Please see our responses in the attached file.

Reviewer 2 Report

This review summarizes the importance of Theragnostic as promising approach to develop accurate in vitro models that reflect the in vivo conditions and developed precise and individualized cure to different diseases. In particular, the focus of the review is the importance of redox homeostasis and mitochondrial function in the context of personalized theragnostic approaches. This is a well written and comprehensive review that explains in a clear and in-depth way the concept of Theragnostic and how this tool can be applied to the diagnostic of neurodegenerative diseases.  

Author Response

Thank you very much for taking the time to review our manuscript. We greatly appreciate your positive feedback and are thrilled to hear that you found our review to be well-written and comprehensive. Your comments on the importance of redox homeostasis and mitochondrial function in the context of personalized theragnostic approaches are encouraging, and we are glad that we were able to effectively convey this concept. 

Reviewer 3 Report

            Theragnostic is a promising approach in the integration of diagnostics and personalized therapeutics. In the present review, the authors described in very detail the importance of redox homeostasis and mitochondrial function in a living organism. The review is very well prepared, illustrated with one table and three figures. The text is very thoroughly prepared on the bases of the latest literature data, the most of references come from the latest few years.

However, only the last chapter was devoted to given in the title theragnostic.

The minor comments:

            1. In paragraph 2.3 there is information concerning the inadequate repair of DNA. It should be stressed that mitochondrial DNA has a very low repair capacity, so the statement is misleading.

            2. In paragraph 3.3 NADPH oxidase is listed as an antioxidative enzyme, whereas the family of NOX enzymes represents very active ROS producers. Please consider it.

            3. Figure 3. The ROS generation in mitochondria is shown on the level of complex IV. Please correct the picture. The ROS formation by mitochondrial respiratory chain complexes depends on the activity of complexes I and III. The CI generate superoxide radical on the matrix side, whereas CIII has two ROS-generating centres, one on the matrix side and one on the intermembrane side.

       In my opinion, the submitted manuscript describes in very detail the mitochondrial role in physiology and pathology. The manuscript seems to be suitable for publication in Antioxidants after considering some comments listed above.

Author Response

We appreciate the time you took to review our manuscript and for providing us with valuable feedback. Our responses to your comments can be found in the attached file.
